# Race and Racism in Historical Fiction: The Case of Jurji Zaydan's Novels

**Esra Tasdelen**

Department of Languages, The College of Dupage, Glen Ellyn, IL 60137, USA; tasdelene@cod.edu

**Abstract:** This paper analyzes the conceptualization of ideas of race in three historical novels in the fictional work of Jurji Zaydan (1861–1914), a Syrian Christian intellectual who wrote on the Golden Ages of Islamic History through serialized, popular works of historical fiction. In the novels analyzed, Fath al-Andalus (Conquest of Andalusia), Abbasa Ukht al-Rashid (The Caliph's Sister), and al-Amin wa al-Ma'mun (The Caliph's Heirs), Zaydan depicts hierarchies of race that are delineated by certain features and categories, especially within the Abbasid among household slaves, and also centers the conflict within the novels around issues of differences in race and lineage. Zaydān shows the importance of rifts in Islamic history stemming from categorizations and distinctions between Arab and non-Arab, or Arab and Persian, or *mawāli*. The novels also reflect the self-conceptualization of Egyptians in relation to their perceptions of the Sudanese, at a time of the rise of Arab nationalism, in late 19th and early 20th centuries.

**Keywords:** Arabic Literature; race; Nahda; Jurji Zaydan

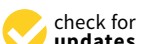



## 1. Introduction

This paper analyzes the conceptualization of ideas of race in three historical novels in the fictional work of Jurji Zaydan (1861–1914), a Syrian Christian intellectual who wrote on the Golden Ages of Islamic History through serialized, popular works of historical fiction at the beginning of the twentieth century. In the novels analyzed, *Fath al-Andalus* (Conquest of Andalusia), *Abbasa Ukht al-Rashid* (The Caliph's Sister) and *Al-Amin wa Al-Ma'mun* (The Caliph's Heirs), Zaydan depicts hierarchies of race that are delineated by certain features and categories. The paper explores how hierarchies of race were utilized by a prominent Arab intellectual in conceptualizing the "other", i.e., the Sudanese, in the Egyptian public sphere at the beginning of the twentieth century.

Jurji Zaydān was one of the most prominent intellectual figures in Egypt at the turn of the century. Zaydān was a direct intellectual product of the Arab Renaissance, the *Nahda*, a series of cultural and literary movements in the Arab world led by Syrian Christians in mostly Egypt and Greater Syria towards the end of the nineteenth century and at the beginning of the twentieth. Most of these intellectuals moved to Egypt in order to avoid the Ottoman Hamidian oppression in the late nineteenth and early twentieth centuries. Once in Cairo, these Christian intellectuals mostly monopolized the press and helped cultivate an increasingly active public sphere.

## 2. The Life of Jūrjī Zaydān

Jūrjī Zaydān was among the Syrian émigrés to Egypt who would later have a profound impact on the Arab cultural sphere and Modern Arabic Literature. Influenced by ideas of social mobility, Zaydān believed that Christian Arabs would have to take a more active role in creating a common and shared national identity for all Arabs, regardless of the region they lived in.

Zaydān's reliance on self-progress and social mobility was a characteristic of Syrian Christians like him. Journalists like Ya'qūb Ṣarrūf (1852–1927) and Jūrjī Zaydān founded

popular Arabic journals such as *Al-Muqtaṭaf* and *Al-Hilāl*. In their journals, the former focused more on Western science, whereas the latter included articles and pieces on literature, society and history.

Zaydān devoted a lot of his time and efforts to the study of history. His work on history was appreciated by many of Zaydān's contemporaries, and according to historians of the early 20th century, Zaydān's historicism was a direct product of the search for and celebration of the nation's origins in the Arab world that marked the end of the nineteenth century and the beginning of the twentieth. This historicism was exclusively harnessed for the cause of distinctive territorial nationalism. The rise of interest in history writing in Egypt in the last decades of the nineteenth century and the first few decades of the twentieth demonstrates this thrust towards historicism.

Zaydān saw himself mainly as an educator of society, and his novels were aimed at familiarizing the common people with their own past in a comprehensible and entertaining manner (Philipp 1979, p. 36). It was Zaydān's goal to arouse a genuine interest in history in the general reader. The press was an invaluable tool at his disposal, and a crucial instrument of expression for the Arab *Nahḍa*.

### 2.1. Fath al-Andalus

Jūrjī Zaydān published his novel *Fatḥ al-Andalus* (The Conquest of Andalusia) in serialized form in his journal *al-Hi*lāl in 1903. The novel is a historical romance set right before the Islamic Conquest of Southern Spain in 711 A.D. In the novel, Zaydān describes the events that led to the Arab conquest of Southern Spain by Tariq b. Ziyad. The narrative is an important tool with which we can understand Zaydān's idea of what constitutes the nation, and a starting point for understanding his later views on Pan-Arab cultural unity.

The major historical event, or the catharsis of the novel, is the conquest of Spain by Ṭāriq ibn Ziyād (670–720), the Berber commander who led the armies of Islam into Spain. With the historical events as the backdrop, the main story is the romance between Florinda (Daughter of Count Julian) and Alfonso (son of Witiza, the late King of Spain). The villain is the tyrant King Roderic, who has usurped Witiza's right to the throne and wants to keep Florinda for himself, within his palace.

In the novel, race is a prominent theme, as Zaydān organizes his characters mainly within a framework of racial categorization, with typical tropes used by racist theory such as facial features, skin and hair color. We see the first example of this in the description of Ṭāriq ibn Ziyād (d.720), the Berber commander who led the Arab armies into Spain, conquered Southern Spain and put an end to the Gothic rule. Racial stereotyping shapes the way that Zaydān describes Tariq when the reader encounters him for the first time:

> Most Berbers have thick lips, broad faces, and short noses. Their hair is normally black, and their skin color is deep brown . . . The general belief was that courage went with brown skin color, while light-skinned people were puny and cowardly. (Zaidan 2010, p. 203)

Tariq, who lives the nomadic life of the Bedouin, is able to integrate into the community regardless of race. The deemphasizing of ethnicity vis-à-vis community is an important concept that also comes up in Muṣṭafā Kāmil's play, *Fatḥ al-Andalus*.

Despite the racial stereotyping, Zaydān downplays the differences between Jews and the Arabs. *Fatḥ al-Andalus* is a historical novel that Zaydān wrote with the idealized concept of a multiethnic, multireligious empire in mind. For Zaydān and his contemporaries, the existence of the Ottoman Empire in the Middle East was still important. Therefore, Zaydān in this novel shows the reader the importance of religious tolerance and just governance. We see this exemplified by the Jewish merchant Sulaymān's words:

> " . . . We Jews are closely related to the Arabs because, as you know, both groups come from a single ancestor, Abraham. They deal with us in a special way. With that in mind, we owe it to them in the current circumstances to support them

in their conquest of this country. In so doing, we're serving our best interests."
(Zaidan 2010, p. 305)

*Fatḥ al-Andalus* is not a typical Zaydān novel, as the protagonists are not Arab, or even Muslim. However, it still gives us important clues regarding Zaydān's intellectual stance on many issues such as governance, gender, religion, minorities and race. As he incorporated intellectual debates of the early twentieth century such as a criticism of female seclusion, minority rights, racial stereotypes and a criticism of religion's involvement with politics into his fiction, Zaydān's novel reads less like a historical novel and more like the reflection of the early twentieth century Egyptian public sphere.

Questions of origins and race in Islam were becoming increasingly important in the times Zaydān lived. At the end of the previous century, Ernest Renan, in criticizing Islam and Arabs in his lecture he gave at Sorbonne in 1883, claimed that Arabs and Persians were different in racial terms. He suggested that both Islam as a religion and the Arabs as a people were hostile to science and philosophy, whereas the Persians, retaining their genius, were an exception among the Islamicized races (Ernest 2000, p. 210). In his famous response to Ernest Renan (1823–1892), the Islamic reformist and political thinker Jamāl al-Dīn al-'Afghānī (1838–1897) employed social Darwinism in claiming that both Christianity and Islam were inimical to science (Al-Afghani 1983). Joseph Massad describes how 'Afghānī undermined Renan's racial comments while criticizing his views:

> Where he disagreed with Renan, however, was on the racialist premises Renan had employed to castigate Arabs as inimical to science and philosophy. Even though he opposed Darwin's theory of evolution in the biological realm, Al-Afghani deployed social Darwinism as the basis of his refutation of Renan. (Al-Afghani 1983) He explained the evolutionary basis of all societies wherein religion, and not "pure reason", emerges in their barbaric state as a transitional phase to civilization. (Massad 2007, p. 13)

Renan thus emphasized the role of Persians living in the Islamic state in preserving and enhancing science and philosophy. Zaydān, in line with this thinking, included many non-Arab/Persian characters such as the Barmakids in his novel *Abbasa Ukht al-Rashid*, and underlined their contributions to the welfare of the state.

### 2.2. Al-'Abbāsa Ukht al-Rashīd

Zaydān's historical novel *Al-'Abbāsa Ukht al-Rashīd aw Nakbat al-Barāmika* (Al-'Abbāsa Sister of al-Rashīd or the Catastrophe that Befell the Barmakids) was published in 1906 in serialized form in his journal *Al-Hilāl*. The novel is set in Baghdad towards the end of the Abbasid caliph Hārūn al-Rashīd's reign, in the year 803 AD. As Issa Boullata has argued, this is a historical novel that uses the setting of the Golden Ages as allegory for the times Zaydān lives in (Zaidan 2011a, p. vii). Together with *Al-Amīn wa al-Ma'mūn* (The Caliph's Heirs), published a year later in 1907 and is in a way the novel's sequel, it gives us a close view of the nature of the Abbasid court and the inner workings of politics and governance as well as the interactions of gender, race, slavery, the debates on lineage and the status of minorities. It is also a classic Zaydān novel, featuring a narrative of romance with political events serving as a background.

The novel is constructed on the power dynamics between Hārūn al-Rashīd, and the Hashemite family/tribe he belongs to on one side, and the Barmakid family, viziers to the Abbasid rulers and coming from Persian origins, thus members of the *Mawālī* (non-Arab clients) on the other. At the center of the novel is the love story between 'Abbāsa, Hārūn al-Rashīd's sister, and Ja'far Al-Barmakī, his vizier. 'Abbāsa and Ja'far have been married to each other by Hārūn al-Rashīd to facilitate his socializing with both of them, yet on the condition that they do not consummate the marriage. However, they both fall in love with each other, and upon learning that 'Abbāsa has two children from Ja'far, Hārūn al-Rashīd orders Ja'far to be executed. This decision is partially due to the court intrigues and the plotting of Ja'far's enemies such as al-Faḍl ibn al-Rabī' who were envious of the close

relationship between the caliph and Ja'far. The execution thus marks the end of an era in which the Barmakids were influential in politics and governance, and of the Golden Age of the Abbasids.

In the novel, the relations between the Arab ruling elite and the *mawlā* (pl. *mawālī*, non-Arab clients) are explored through the interactions between different characters. Zaydān touches on the roles and rights of minorities and different races in an Islamic society by introducing us to non-Arab characters such as Abū al-'Atāhiya, the court poet.

The novel also depicts intricate hierarchies within different strata in the Abbasid society and the harem. When 'Abbāsa, the sister of Harun al-Rashid, complains of her brother denying her the consummation of her marriage to a *mawlā* (Ja'far), she questions the hierarchy and asks 'Utba:

> If my brother really considers marriage to clients or slaves to be a lowering of the station of the caliphate, why then does he marry slave-girls and sire children from them whom he appoints as heirs apparent? Or is a slave-girl higher in rank than a client? (Zaidan 2011b, p. 19)

In the Abbasid social pyramid, as the ruling elite considers itself to be from a purely Hashemite lineage, both the *mawālī* and the slave girls are considered to be much lower in rank than the Arabs. However, the exact hierarchy within these lower strata of society is fluid. Some *mawālī* (like Ja'far al-Barmakī or Abū al-'Atāhiya) rise to the ranks of statesmen or court poets in the caliph's ruling circles, and some slaves are able to rise within the hierarchy by marrying a caliph or other members of the elite and being freed. An example to this is Ḥamdān, Ja'far's slave boy who has been with him since his childhood:

> Hamdan was now fifty years old, but he was still nimble and active. He was a Persian from Khurasan and was a favorite of Ja'far; he could enter into his presence whenever he wanted, and Ja'far treated him like a member of the family. (Zaidan 2011a, p. 49)

Throughout the novel Zaydān continuously contrasts 'Abbāsa's race with that of Ja'far's, and reminds the reader constantly that the lovers cannot be together due to their different racial backgrounds. This gap presents the biggest challenge to Ja'far and 'Abbāsa's love affair:

> In spite of Abbasa's love for Ja'far and her wholehearted dedication to his happiness and comfort, he could not forget that her lineage, as was clear to all people of that age, was considered to be more honorable than his. For she was a Hashemite Arab, the daughter of the caliph, and the sister of another caliph, while he was a Persian and a foreigner who, despite the status and power he had achieved, was still considered a mere *mawla*, or a foreign member of an Arab tribe. Since the beginning of Islam and until this time, no person among the non-Arabs, whatever power or high rank he had achieved, could aspire to any position similar to the one which Ja'far had reached–not even kings and potentates. (Zaidan 2011a, p. 54)

Ja'far is an exception to the rule of segregation between the Arabs and the non-Arab clients, yet he is nevertheless banned from consummating his marriage to 'Abbāsa. In the novel, Ja'far repeatedly reminds 'Abbāsa of his racial lineage:

> Destiny has ordained that we should both suffer, because it has placed a curtain of honorable lineage between you and me: it made you one of the honored descendants of the tribe of Hashem, Prophet Muhammad's folks, and made me a *mawla*, one of the non-Arab clients. (Zaidan 2011a, p. 55)

'Abbāsa, however, believes that Ja'far is higher in esteem than the whole tribe of Hashem. Yet Ja'far prohibits her from addressing him as "my master":

> Don't say 'my master' to me. I am your *mawla*, and you are my mistress, according to their law and tradition. Who am I compared with the sister of the Commander of the Faithful? (Zaidan 2011a, p. 61)

The novel is crucial to discussions of race in Arabic fiction, as the major conflict in the novel is a racial one: Zubayda (Hārūn al-Rashīd's wife and Al-Amīn's mother) and Al-Amīn form the Hashemite side of the two camps, and Ja'far and Al-Ma'mūn are on the opposite side as non-Arabs, and those closer to the followers of Ali. Al-Ma'mūn as a child has been educated by Ja'far al-Barmakī, and therefore is sympathetic to the Persians and their cause. Also on the Hashemite side is Ja'far ibn al-Hādī, the son of Hārūn al-Rashīd's brother, Al-Hādī, who lost his claim to the throne. Ja'far claims that the *mawālī*, or the non-Arab clients have usurped his right to the throne and taken the caliphate from him.

This racial conflict is also evident in physical descriptions of racial features, in a way very similar to what we have seen earlier with the novel *Fath al-Andalus.* Purity in race is once more attached to skin tone, and in Zaydān's descriptions of Zubayda, both her physical features and her character are tied to her race:

> Zubayda, daughter of Ja'far ibn Abu Ja'far al-Mansur, was the cousin of al-Rashid and his wife, whom he married in the year 165 AH (781 AD). Zubayda had first place with Al-Rashid, who preferred her to his other wives because of her Hashemite lineage and beauty. (Zaidan 2011a, p. 109)

> Zubayda had a bright face and was of white complexion. She possessed the dignity of the Hashemites, with a certain sweetness and beauty. (Zaidan 2011a, p. 112)

> Even 'Abbāsa's two sons Al-Ḥasan and Al-Ḥusayn reflect the characteristics of their Hashemite lineage, manner of speaking and eloquence. Zubayda is especially devoted to the Hashemites and harbors a grudge against the Barmakids (and especially Ja'far) as they have helped Al-Ma'mūn become a successor to the throne, discrediting Al-Amīn in the process. In his *History*, Zaydān underlines the fact that most of Ja'far's enemies in the caliph's palace were Arabs or those connected with Arab families. Quoting the historian Mas'ūdī, Zaydān relates that out of the enemies of Ja'far al-Barmakī, the most ill disposed towards him and the one who had the ability to injure him the most was Zubayda, Al-Amīn's mother, as he had irritated her by preferring her rival's son. (Zaydān 1907, p. 196)

Zaydān also describes in detail the special rank that Hashemites held in the Abbasid Empire:

> According to the conventions of those days, the Hashemites used to be called "Sons of Kings" or "Nobles" (Sharifs). (Zaidan 2011b, p. 123)

> Zaydān emphasizes this special status repeatedly, especially when Ismail ibn Yahyā and Hārūn al-Rashīd are speaking about Ja'far ibn al-Hādī, who is also a Hashemite. Hārūn al-Rashīd states, referring to him: "Relationship to the Messenger, Prophet Muhammad, is the greatest source of honor for him and for us." (Zaidan 2011b, p. 128) Zaydān repeatedly provides evidence for the special status of the Hashemites, such as the hierarchy reflected in the construction of the palace and its different sections. When Ja'far is summoned to Hārūn al-Rashīd's palace, his horsemen cannot move further than the fourth gate in the palace grounds, "which was accessible only to him and the Hashemites and similar elites." (Zaidan 2011b, p. 153)

Towards the end of the novel, the conspirators against Ja'far al Barmakī and his family claim that the Barmakids have acquired too much power as non-Arabs, and in an unprecedented way. When Hārūn al-Rashīd is convinced that the Barmakids have ulterior motives, he asks Ismā'īl:

> Don't you think these non-Arabs have become presumptuous towards us and have taken exclusive possession of the state and its funds to our detriment? (Zaidan 2011b, p. 161)

The substitution of the word "Non-Arabs" for "Barmakids" is crucial here, and towards the end of the novel the Hashemites increasingly refer to the "non-Arab" status of

the Barmakids, and connect all the accusations towards them to their lineage. None of the two sides are immune to utilizing racial stereotypes to further the conflict: Ja'far, while talking about Hārūn al-Rashīd, and is afraid that the "greed of the Hashemites" will show itself in him.

'Abbāsa's relationship with Ja'far, when it is revealed to Hārūn al-Rashīd, is in his eyes an unforgivable sin and betrayal. Hārūn al-Rashīd, as he is about to have 'Abbāsa executed, asks her:

> Does someone like you betray her brother with a man from the *mawali*?
>
> (Zaidan 2011b, p. 186)

Once more, the betrayal of 'Abbāsa takes on the characteristics of a racial conflict, as she has preferred partnership with a non-Arab client to all the other Hashemites. Furthermore, Hārūn al-Rashīd's resentment towards Ja'far al-Barmakī increases as he believes that Ja'far's union with 'Abbāsa is a political maneuver, aimed at raising Ja'far's rank as a non-Arab:

> This was especially so because he thought that his vizier had only sought to have children from al-'Abbāsa so that his sons would have Hashemite blood. That would entitle him to assume the caliphate, which was limited to Qurayshi descendants in those days. (Zaidan 2011b, p. 189)

Therefore, we can claim that in *Al-'Abbāsa Ukht al-Rashīd*, the major conflict is a racial one, stemming from a major challenge to the social hierarchy between different races. 'Abbāsa and Ja'far al-Barmakī challenge the rules governing Arab-non-Arab relations, but they pay the ultimate price, and in the end their family is destroyed.

For Zaydān, the service of the Barmakids to the state was more important than their lineage. By focusing on lineage, Zaydān suggests that the privileges given to the Hashemite elites and the strong emphasis on race and lineage were part of the reasons for the decline of the Abbasids.

The issue of lineage was important in the period in which Zaydān was writing. Towards the end of the nineteenth century and at the beginning of the twentieth, Arabs were demanding cultural rights and language rights within the multilingual, multiethnic Ottoman Empire. Zaydān, by underlining the importance of service to the state and not ethnicity, may be alluding to the exclusivity of proto- Turkish nationalism increasingly prevalent within the Young Turk circles, and the Hashemite lineage may well have been an allegory for Ottoman/Turkish lineage. Just as the Barmakids, who were non-Arabs, nevertheless contributed to the Abbasid state, Zaydān is suggesting that Arabs, as non-Turks, can also contribute to the Ottoman state, as long as they are granted their rights in the areas of language and culture.

In addition to female characters and themes of gender, the novel also includes detailed descriptions of slavery, the slaves in the harem and the hierarchy between them. Early in the novel, Zaydān introduces the reader to Phinehas, a wealthy slave merchant in Baghdad, and shows us how slave trade in the Abbasid Empire was a common and profitable practice. Phinehas is described as one of the wealthiest people in the city, having amassed his fortune from the sale of slaves to the caliphs, their children, the viziers, and other elites.

Phinehas owns a very wide variety of slaves, and the diversity in skin color (white, yellow, red and black) as well as races (Slavic, Byzantine, Turkish, Persian, Armenian, Sindhi or Berber) in the group of slaves generates a hierarchy. The ideal slave is one who has memorized delightful poems and can play musical instruments perfectly. Racial hierarchy based on skin tone once more governs the hierarchy and order: White slave girls are preferred to ones with darker skin, as they are considered to be more beautiful (Zaidan 2011a, p. 31).

The description of slaves and different slave groups in the fictional narrative is very similar to Zaydān's depiction of slavery in the Abbasid Empire in his non-fictional, historiographical work, *History of the Islamic Civilization*. In this work, Zaydān describes how with the spread of Islam and the Arab conquests, groups of slave-women were acquired.

Zaydān adds that some of these women were employed as nurses, wet and dry, and the younger ones were made concubines (Zaydān 1907, p. 211).

The racial hierarchy among the female slaves is evident in conceptions of beauty, and Zaydān suggests that the black slaves are considered to be "closer to a wild and squalid desert state" than other girls with darker skins. (Zaidan 2011a, p. 33) Zaydān in the original text uses the term *banāt al-zunūj* (black girls) for these slaves, and describes them as "little girls with black skin, curly hair, and flat noses", with classic racial stereotyping. (Zaidan 2011a, p. 33) Inspired by the work of Eve M Trout Powell, I claim that just as Egyptians at the beginning of the twentieth century were drawing images of the "other", i.e., Sudan and the Sudanese, in stereotypes within the public sphere, Zaydān was describing black slaves in the Abbasid palace in similar terms. The racial hierarchy based on skin color among the slaves in the harem is representative of " . . . the hierarchy of color that existed in the army and in the levels of authority . . . Circassian children . . . . were considered to be more valuable, and cost more than darker children." (Powell 2003, p. 91).

There are some historical examples of this racial stereotyping and the hierarchy within the system of slavery. Bernard Lewis traces this hierarchy back to the first centuries of Islam, and states that groups of black Africans were lumped together under the general heading "Sudan": "The Zanj are the least respected, the Ethiopians the most". (Lewis 1971, pp. 30–31) Indeed, Mas'ūdī, the author of one of the historical sources that Zaydān uses in the sequel *Al-Amīn wa al-Ma'mūn*, has written lengthy depictions of "the black man", attributing to him "frizzy hair, thin eyebrows, broad nostrils, thick lips, pointed teeth, smelly skin, black eyes, furrowed hands and feet . . . " (Lewis 1971, p. 34).

In line with the Orientalist and colonialist rhetoric of the end of the 19th and the beginning of the 20th century, Zaydān provides us with binaries of 'civilized' versus 'rugged/simple', or 'city' versus 'the desert'. He points out to the necessity of "urban civilization" to educate the "nomadic, pure, innocent" girls through the words of the slave merchant Phinehas:

> Being enslaved is one of the most important causes of their happiness, for they move from the coarse life of the desert and its rugged existence to the city and its luxuries. (Zaidan 2011a, p. 33)

This "white man's burden" of civilizing the slaves was a trope very common in the attitudes towards the slaves brought especially from Africa, throughout Islamic history. Bernard Lewis suggests that the Islamic lands viewed both "the white barbarians of the North" (Turks, Slavs . . . etc.) and "the dark-skinned barbarians of the south" (African blacks) were perceived as potential slaves to be imported into the Islamic world (Lewis 1971, pp. 28–29).

Slavery within the harem was a big part of the social reality of the Ottoman palace. Orit Bashkin, while analyzing Zaydān's elaborations on the harem, connects them to the Ottoman views of slavery prevalent at the time: "Within this debate, a "rags-to-riches" narrative was employed in defense of the harem system, as the harem elevated female slaves from their lowly rank to the top of the imperial system." (Bashkin 2010, p. 299).

Powell has shown in her work on slavery in Egypt towards the end of the nineteenth century that slavery "was part of an ages-old trade relationship between Egypt and the Sudan that symbolized the special connection between the two countries, a connection often framed as a domestic exercise in a national "civilizing mission"" (Powell 2003, p. 3).

The racial stereotyping regarding the imagery of Sudanese slaves and Sudan would continue to permeate the Egyptian press even in the later decades, the 1920s and even the 30s. Beth Baron shows us how in the satirical political journal *Al Kashkūl* (The Scrapbook), the female figure representing Egypt "almost always appears with a fair complexion, whether she is a young girl, 'new woman', or peasant. By comparison, the Sudan–depicted as a nearly naked woman and in other guises–has darkened skin and exaggerated facial features. The Egypt of *al-Kashkul* is clearly not akin to territories to the south and of the African continent but closer in resemblance and style to the European neighbors to the north." (Baron 2005, p. 119).

The Caliph's Sister gives us important clues as to where racial tensions lay in Zaydān's conceptualization of Islamic history, as well as the delineation of racial hierarchies based on skin tone and physical features of slaves in the Abbasid harem. The categorization of those with darker skin is representative of the historical context Zaydān operated in, with the Egyptians perceiving the Sudanese as the "other", especially at the beginning of the 20th century.

### 2.3. *Al-Amīn wa al-Ma'mūn: The Caliph's Heirs*

Zaydān's sequel to *Al-'Abbāsa Ukht al-Rashīd* (The Caliph's Sister) is *Al-Amīn wa al-Ma'mūn* (The Caliph's Heirs), published in 1907. The novel picks up the historical timeline at the point where its prequel, *'Abbāsa*, ends, and describes the political struggles for succession to the Abbasid throne as a background to the romance between the two main characters: Maymūna (The daughter of Ja'far al-Barmakī) and Behzād (a doctor of Persian origin from Khurasān, and as is later revealed, the great grandson of Abū Muslim al-Khurasāni, who was executed after the Abbasid state was founded).

The major challenge and the conflict in the novel, once again, stems from differences in race and lineage. In his analysis of the novel, Michael Cooperson notes that Zaydān's casting of this conflict in ethnic terms (i.e., "the Persians" versus "the Arabs") was characteristic of historians in the early twentieth century in general. (Zaidan 2011b, p. x) The novel focuses on the struggle for succession between the two brothers Al-Amīn and Al-Ma'mūn, who come from different lineages. As in *Al-'Abbāsa Ukht al-Rashīd*, racial categorization creates the major conflict in the novel: Zaydān emphasizes the advantage of the Hashemite lineage of Al-Amīn compared to that of Al-Ma'mūn, the son of a Persian slave-girl. In the novel we witness how the civil war that broke out between the two heirs paves the way to the decline of Abbasid central power and the eventual death of Al-Amīn in the hands of Al-Ma'mūn. Throughout the novel, the lovers Behzād and Maymūna desire not only a union, but also the revenge for the unjust death of Abū Muslim al-Khurasāni.

Slaves, once again, are a big part of the household in this novel as well. Zaydān describes the upbringing of the children of elite families by slaves as the norm of the period:

> In those days, parents rarely lived with their children, preferring to place them in the care of slave women. So it was that Zaynab was raised to be a philosopher, indifferent to all but the facts of the case, and scornful of the frolicking and merriment that captivated her peers. (Zaidan 2011a, p. 29)

The stereotypes that Zaydān employs to describe slave boys and girls and their place within the harem suggest that this phenomenon of the children of the household being entrusted to slaves for their upbringing seems to be directly inspired by the relation between Egyptian nationalists and the Sudanese slaves in their households, as described by Powell:

> Egyptian nationalist activists who grew up in large households in the upper echelons of society came of age under the care of domestic slave and servants, the majority of whom were Sudanese. Slaves would have thus been part of nationalists' pronounced sense of home, and not only the physical household but also the traditions of family structure that bound these Sudanese servants to them. (Powell 2003, p. 181)

Slaves within the household were mostly of African origin: Bernard Lewis points out that whereas white slaves were imported into the Islamic empires to form the main corpus of the military (such as the example of the janissaries in the Ottoman Empire in later periods), "The main purpose for which blacks were imported was domestic service." (Lewis 1971, pp. 81–82).[1]

The racial hierarchy is apparent in the fictional universe of this novel as well. Al-Ma'mūn's mother, Marājil, a Persian slave girl, has a lower status in the hierarchy. Zubayda, the mother of Al-Amīn who comes from a purely Hashemite lineage, humiliates Al-Ma'mūn by calling him not by his name but by the derisive nickname "The Son of Marajil"

(Zaidan 2011a, p. 202) Bernard Lewis traces the contempt of the slave mothers and their offspring to the early centuries of Islam and even to pre-Islamic times:

> In early Islamic and pre-Islamic times the Arabs looked down on the sons of slave-mothers, regarding them as inferior to the sons of free-born Arab mothers. The stigma was attached to the status, not the race of the mother, and affected the sons of white as well as black concubines. Before long, however, a distinctive color prejudice appeared, and the association of blackness with slavery, and whiteness with freedom and nobility became common. (Lewis 1971, p. 94)

The strong enmity between Zubayda and Al-Ma'mūn's mother was rooted in existing power dynamics in Islamic history and not solely a product of Zaydān's imagination. In his *History of Islamic Civilization*, Zaydān describes and references this hatred and jealousy mutually felt by Zubayda and Al-Ma'mūn's mother. (Zaydān 1907, p. 184) Tayeb El-Hibri's work on Islamic historiography elaborates on the racial hierarchy in the Abbasid Empire:

> As mother of Al-Amin, and only wife of Harun, Zubayda found it inconceivable that the caliph might place Al-Ma'mun, the son of a concubine, before her son in the order of succession. Ignoring Al-Amin's flaws she therefore pushed to secure the succession for him. (El-Hibri 1999, p. 43)

Towards the end of the nineteenth century, the presence of slaves within the Egyptian household was complicated by the rise of nationalism and the perceptions of Egyptian self-identity. Eve Trout Powell describes how towards the end of the nineteenth century with the writings of intellectuals such as Qāsim Amīn, the Egyptian household became a metaphor for Egypt's engagement with modernity, and therefore the presence of domestic servants and slaves had to be addressed (Powell 2003, p. 3). This might have inspired Zaydān to include the detailed descriptions of slaves and their roles in the Abbasid harem of his work of historical fiction. The constant presence of especially black slaves in Zaydān's two novels *Al-'Abbāsa Ukht al-Rashīd* and *Al-Amīn wa al-Ma'mūn* was necessary in exactly the same way that " . . . the presence of the Sudanese and the Nubians in Egypt during the late nineteenth century was crucial for the development of Egyptian national identity and for situating where Egyptians stood culturally in relation to Africa and Europe." (Powell 2003, p. 70). Thus, Egyptian national identity was formed while situating itself in relation to an "other" within Egyptian households. Powell describes this phenomenon of "the colonized colonizer" in her analysis of different intellectual products of the national imagination.

In the racial hierarchy of the Abbasid harem of the novel, then, freemen and women are above slaves, and Arabs are above non-Arabs. As we have seen in the prequel *Al-'Abbāsa Ukht al-Rashīd*, within the slaves there is a hierarchy as well: White slaves are valued more than black slaves. For Lewis, this hierarchy and different levels of social mobility can be traced back to the early Islamic Empires:

> Whereas white slaves could become generals, provincial governors, sovereigns, and founders of dynasties, this hardly ever happened with black slaves in the central Islamic lands. (Lewis 1971, p. 78).

As we have seen above, in the novel Zaydān casts the succession conflict as one between "The Arabs" and "The Persians". Al-Amīn's status as the only heir whose mother and father are both of Hashemite origins gives him an exclusive advantage. Zubayda, Al-Amīn's mother, when addressing Al-Amīn, emphasizes this:

> You're the only caliph born to Hashimite parents on both sides, and your children have a better pedigree than the rest of the 'Abbasids. (Zaidan 2011b, p. 253)

Even Zaynab, the daughter of Al-Ma'mūn, who is one of the most intelligent and sophisticated characters in the novel, boasts of her Hashemite lineage:

> Like her grandfather al-Rashid, she was a fierce partisan of her clan, the House of Hashim . . . She was often present during his conversations with his wife

Zubayda, who frequently boasted of her Hashimite lineage. Overhearing their conversations, Zaynab came effortlessly to acquire the same pride in her clan, despite Danānīr's efforts to root out any such chauvinism. (Zaidan 2011a, p. 25)[2]

Zaynab's partisanship for the House of Hāshim is very similar to a national pride that has priority over other loyalties and responsibilities. The senior members of the Hashemite family are physically very close to the caliph and reside in Baghdad, and are also present in the oath of allegiance ceremony (Zaidan 2011b, p. 83).

Despite the fact that the Hashemites were so proud of their lineage, in Islamic history, the ruling elites became increasingly mixed with the peoples they conquered. Zaydān elaborates on the issue of lineage in his *History of Islamic Civilization* and suggests that the "purity" of the race was simply a fantasy:

> Arabian pedigrees were kept pure only in pagan days, and at the beginning of Islam, to the middle of the Umayyad period: after that purity was confined to the father's side; on the mother's side they became exceedingly mixed. (Zaydān 1907, p. 214)

The most obvious evidence of this "mixed pedigree" is Al-Ma'mūn, who is the son of a slave-girl but has an equally legitimate claim to the throne.

### 3. Conclusions

In this paper I have analyzed three examples of historical fiction composed by Syrian Christian author Jurji Zaydān: *Fath al-Andalus, Al-'Abbāsa Ukht al-Rashīd* and *Al-Amīn wa al-Ma'mūn/*. These novels show us that the historical fiction writing of the early 20th century dealt with important themes such as race, gender and lineage. By exploring the major themes in the novels as they relate to the issue of race and racism in Islamic history, the paper has explored the connections between Zaydān's fiction and the context of the Egyptian public sphere at the beginning of the 20th century. The novels depict racial hierarchy in the Abbāsid harem and among household slaves as reflective of the same hierarchical system in several Islamic empires. Allusions to key stereotypes of race such as tones of skin color determine a slave's placement within the hierarchy. Another prominent theme in Zaydān's historical novels is that their central conflict is built on issues of differences in race and lineage. Zaydān shows the importance of rifts in Islamic history stemming from categorizations and distinctions between Arab and non-Arab, or Arab and Persian, or *mawāli*. A final point that the paper explored is how Zaydān's novels reflect the self-conceptualization of Egyptians in relation to their perceptions of the Sudanese, at a time of the rise of Arab nationalism, in late 19th and early 20th centuries. Historical fiction disseminated by the printing press was a powerful tool in Cairo, and hierarchies of race were utilized by a prominent Arab intellectual such as Zaydān in conceptualizing the "other", i.e., the Sudanese, in the Egyptian public sphere at the beginning of the twentieth century.

**Funding:** This research received no external funding.

**Conflicts of Interest:** The author declares no conflict of interest.

### Notes

[1]　Lewis adds that "Black slaves for domestic use were very common during the nineteenth century in Egypt, in Turkey and in the other Ottoman lands, and some survivors can still be met in these countries . . . The Nubian porter, servant or hawker remains a familiar figure in Egypt to this day." The timing is important for our analysis as it directly precedes Zaydan's lifetime.

[2]　Zaydān uses the word '*Aṣabiyya*' which Michael Cooperson translates as 'chauvinism'.

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
