# Peer review of "Race and Racism in Historical Fiction: The Case of Jurji Zaydan’s Novels"

_humanities, doi:10.3390/h10040119_

Round 1
Reviewer 1 Report
Race and Racism in Historical Fiction: The case of Jurji Zaydan’s novels
In this article the author argues that Jurji Zaydan discusses race and racism in the three novels they examine. The author zeros in on the hierarchy that characterizes Arab Dynasties Abbasids). Through their analysis of the novels, the writer provides examples to show how the “Other” is marginalized in the court culture of the Abbasids, illustrating how racial discrimination permeates court culture. While I find the author’s selected passages from the novels very compelling, the analysis provided is deficient, and it should be expanded to bolster the writer’s argument (s). Also, while it is very commendable to provide a long biography of Jurji Zaydan, it is not necessary in a scholarly article; a footnote could be sufficient. Furthermore, a bit more tinkering with the theoretical apparatus, engagement with existing scholarship and theory, and better signposting of the argument would go a long way to helping the readers to understand precisely what the author is attempting to do in this piece.
In addition to these issues, there are also a few stylistic issues, and I will list some of them below:
- Footnotes need more work.
- Issues with quotations (formatting issues)
- Indentation
- Grammatical issues
- Repeated words
- Lack of engagement with theory on race
- No Keywords
- Abstract needs to be rewritten.
This piece has to be completely revised before it gets published, and I suggest that the author does extensive reading on race to enrich his analysis and bolster his argument. Also, the sources consulted shouldn’t be limited to Bernard Lewis and Eve Powell.
Author Response
The paper has been revised with more analysis of the themes, and I added a new abstract and conclusion that brings together the key points that the paper looks at throughout the analysis of the novels. Language, grammar and indentation issues have been resolved. Keywords have been added.

Reviewer 2 Report
My main concern with the paper is the lack of a clearly articulated argument. The paper is about race, power, gender, among other things. The author discusses all of these issues, but the reader will wonder what the glue that brings all these issues together is. I made specific suggestions, which I embedded into the PDF as comments. Please see them in the attached document.
Another concern is the timidity with which judgements of the discussed issues is expressed. The author often gives "descriptive" statements (like X thinks that ... or Y says this ....) without attempting to take sides, approve or condemn the statements, or provide the author's own take on the issues. This is what rigorous academic writing is all about, and this was absent from the paper. I embedded detailed comments into the document. Please see them in the attached file.
The paper is good, overall, but the Introduction and the Conclusion in particular need to be revised. I commend the author on his/her very accurate use of transliteration symbols (only one mistake that I pointed out in the embedded comments). This is very rare, and it made the reading experience very pleasant.
Finally a round of proofreading is needed to fix style, grammar, and punctuation issues.

Author Response

(The authors gave the same response as above.)

Reviewer 3 Report
Dear authors,
I would consider expanding the introduction a bit more. At the end of it, I would include a paragraph where I would give the reader a preview of the structure of the article.
I would propose the analysis of each of the novels in a separate section (3), and not in the second section, where the author's biographical details are presented first.
In the analysis offered on the interpretation of the concept of race in the work of this author, it would be interesting to point out other works (scientific or literary) that would serve to contrast, expand or even debate the assumptions of this particular author. Otherwise, it seems that this proposal is limited to a purely descriptive exercise, and does not take into account the importance of a deeper analysis in relative terms.
It would be useful to draw more extensive conclusions. In them, I would offer a review of what has been mentioned previously and, in addition, I would include original proposals that go beyond what has already been said in the analysis offered previously.
It would be convenient to review the rules of citation for the specific journal. There are no references at the end of the article, and footnotes do not seem to be the norm for this journal.
Author Response

(The authors gave the same response as above.)
